# Human Activity Recognition for the Identification of Bullying and Cyberbullying Using Smartphone Sensors

**Vincenzo Gattulli** \* , **Donato Impedovo** , **Giuseppe Pirlo and Lucia Sarcinella**

Dipartimento di Informatica, Università degli Studi di Bari Aldo Moro, 70125 Bari, Italy
\* Correspondence: vincenzo.gattulli@uniba.it; Tel.: +39-32140455

**Abstract:** The smartphone is an excellent source of data; it is possible to extrapolate smartphone sensor values and, through Machine Learning approaches, perform anomaly detection analysis characterized by human behavior. This work exploits Human Activity Recognition (HAR) models and techniques to identify human activity performed while filling out a questionnaire via a smartphone application, which aims to classify users as Bullying, Cyberbullying, Victims of Bullying, and Victims of Cyberbullying. The purpose of the work is to discuss a new smartphone methodology that combines the final label elicited from the cyberbullying/bullying questionnaire (Bully, Cyberbully, Bullying Victim, and Cyberbullying Victim) and the human activity performed (Human Activity Recognition) while the individual fills out the questionnaire. The paper starts with a state-of-the-art analysis of HAR to arrive at the design of a model that could recognize everyday life actions and discriminate them from actions resulting from alleged bullying activities. Five activities were considered for recognition: Walking, Jumping, Sitting, Running and Falling. The best HAR activity identification model then is applied to the Dataset derived from the "Smartphone Questionnaire Application" experiment to perform the analysis previously described.

**Keywords:** human activity recognition; deep learning; machine learning; smartphone; bullying; cyberbullying

## 1. Introduction

*Human Activity Recognition (HAR) systems* are concerned with the recognition of activities performed by humans [1]. The activities are among the most varied, e.g., walking, sitting, falling, etc. The activities are performed by individuals generally wearing positioned devices such as smartphones and smartwatches. Devices such as smartphones or smartwatches monitor the physical and psychological states of users. These devices implement sensor hardware and through them can extract new information. This can be through inertial measurement systems (or Inertial Measurement Units, IMUs), systems used in aviation to study the movements of an aircraft, or more simply through sensors found on everyday devices such as smartwatches or, indeed, smartphones. Depending on the sensors used, we speak of n-axis IMUs. If, for example, one includes an accelerometer (three axes) and a gyroscope (three axes), one speaks of a six-axis IMU. If one also includes the magnetometer, one speaks of a nine-axis IMU. The sampling frequency of the sensors used is relative to the activity of interest and ranges from tens of Hz to several hundred.

Among the various sensors, the triaxial accelerometer is the most widely used sensor in smartphone devices, but other sensors such as the triaxial gyroscope, magnetometer, and many others can also be found. Various research studies have found that:

1. The accelerometer detects linear motion and gravitational force by measuring acceleration in the three axes X, Y, and Z;
2. The gyroscope is used to measure the rate of rotation of a body;

3. The magnetometer, rather, is used to detect and measure geomagnetic fields. The latter is not always used for HAR purposes, so it is not always included in the data analyzed.

This work discusses a new smartphone methodology that combines the final label elicited from the cyberbullying/bullying questionnaire (*Bully*, *Cyberbully*, *Bullying Victim*, and *Cyberbullying Victim*) and the human activity performed (*Human Activity Recognition*) while the individual fills out the questionnaire. In addition, for the scientific aspect, a new Dataset named "*DatasetUniba*" inherent to Human Activity Recognition (containing the following activities: walking, running, jumping, and sitting, with, in addition, and contrast to other scientific papers, "falling") will be introduced. The problem focuses strongly on finding an existing behavioral correlation between the activity/behavior and the individual's personality index (label), while they fill out a simple questionnaire dealing with sensitive social issues. Therefore, HAR methodology was used differently from other papers, which only try to discriminate activities in the dataset they consider as best they can.

The paper highlights these methodological aspects:

- In this paper, we want to perform activity recognition i.e., (walking, running, jumping, sitting, and falling). Action recognition in Human Activity Recognition is treated very much as state of the art. In particular, in this paper, falling is also recognized, which is not considered in many papers. Compared to the state of the art, the recognition of the primary actions of Walking, Running, Jumping, and Sitting is in line with the state of the art discussed in this paper.
- For action recognition, a dataset "DatasetUNiBA" that detects actions (walking, running, jumping, sitting, falling) was created by our research team in the lab. This dataset was compared with a publically available dataset, UniMiB SHAR, which is a state-of-the-art dataset for Deep Learning algorithms, and among the best known in the literature.
- After identifying the best model for Human Activity Recognition, a new smartphone methodology was proposed that combines the final label (personality index) triggered by the cyberbullying/bullying questionnaire (Bully, Cyberbully, Bullying Victim, and Cyberbullying Victim) and the human activity performed (Human Activity Recognition) while the individual fills out the questionnaire. In practice, this smartphone methodology elicited accelerometer data acquisition while 52 users for test 1 and 48 users for test 2 filled out a questionnaire drafted by psychologists.

The submitted questionnaire deals with questions about the active and passive actions of bullying and cyberbullying drafted by psychologists. Through the final labels of the questionnaire, i.e., "Bully, Cyberbully, Victim of Bullying, and Victim of Cyberbullying", conclusions were drawn by comparing the received label of the questionnaire with the activity performed by the user. Restless behavior, other than just sitting, can characterize an identifying behavior for the category. As the individual fills out the questionnaire, the app will extrapolate the sensors, specifically the accelerometer. A normal questionnaire could not capture users' movements to extract new behavioral knowledge.

The paper is structured as follows: Section 2 summarizes some current approaches inherent to Human Activity Recognition with particular attention to the Smartphone-based category; Section 3 explains the Datasets used in the experimentation phase; Section 4 describes the Data preprocessing and models and then the design of the experiment; Section 5 illustrates the results and analysis of the Dataset extracted from the real-world test with the questionnaire application. Finally, Section 6 analyzes the results and concludes by discussing strengths and weaknesses.

## 2. State of the Art

This chapter reviews the most interesting studies in this field. Our research team deals with projects related to the topic of Cyberbullying and Machine learning. Specific papers dealing with Human Activity Recognition with Cyberbullying are few. For this



reason, we discuss papers strongly related to the concept of Human Activity Recognition but not inherently to Cyberbullying. The Human Activity Recognition approach is used in this paper for the first time to identify possible activity-related anomalies while taking a psychological questionnaire, which could identify the individual's attitude inherent in bullying/cyberbullying.

Many studies do not adopt a Feature Extraction and Feature Selection phase; the following are some examples of this type of study. The study by Minarno et al. [2] recognizes the daily life actions of thirty volunteers using a triaxial accelerometer and gyroscope. Minarno et al. [2] test various classical machine learning classifiers, such as Decision Trees, Random Forest, and K-Nearest Neighbor, on a set of daily life actions consisting of lying down, standing, sitting, walking, and going down- or upstairs. Without implementing a feature selection and extraction step some of the classifiers perform very well, with up to 98% accuracy.

In the study by Cho et al. [3] both approaches are tried, using only accelerometer data. Three public datasets are chosen, *UMAFall* [4], *UniMiB SHAR* [5], and *SisFall*. The latter is a publicly available dataset not usually considered because it uses only Inertial Measurement Units to collect data. Of these three datasets, Cho et al. perform a Leave One Subject Out Cross Validation using a Convolutional Neural Network as the classifier, but with various features. They try various combinations formed between raw data, *Signal Vector Magnitude*, *Singular Value Decomposition*, *Kernel Principal Component Analysis*, and *Sparse Principal Component Analysis*. The results show that the use of Signal Vector Magnitude [6] alone leads to severe feature lowering but, by accompanying it with the raw signal features, this increases its performance, at least in the case of UMAFall and UniMiB. The maximum achieved in these experiments is obtained by UniMiB, with 75.65% accuracy. Conversely, the accuracy with UMAFall is 64.69% in the best combination.

The study by Concone et al. [7] aims to recognize only four actions with data from the accelerometer and gyroscope. The actions are standing still, driving a vehicle, walking, and running. The sliding window used is three seconds without overlap. This study aims to compare its results obtained with those of most frameworks and the Google API. The best-performing classifier turns out to be a K-Nearest Neighbor with an accuracy of 95.43%, precision of 91.98%, and recall of 92.87%. For the other classifiers, there is difficulty in distinguishing between standing still and driving a vehicle due to the nature of the acceleration values. The Google API, according to the results of this study, would have a hard time distinguishing between walking and running.

The approach proposed by Gupta [8] in his study is one without feature selection and extraction in that a Convolutional Neural Network and a Deep Convolutional Long Short-Term Memory are again chosen as classifiers, which, like all Deep Learning algorithms, can compute a set of features of interest by itself to allow for a smoother final classification step. In this study, Gupta tried to distinguish among the activities available in the WISDM dataset [9].

These activities are divided, for classification convenience, into three categories:

- Ambulation-Oriented Activities;
- Hand-Oriented Activities (General);
- Hand-Oriented Activities (Concerning Food).

The first activities include actions involving body movements, thus walking, climbing stairs, standing, jogging, kicking, and sitting. These actions are recognized using WISDM data that predicted that the smartphone was positioned at the waist. For the other two types of actions, which involve movements that affect only the subject's hands and not his or her entire body, data are obtained through the use of a smartwatch.

In the case of the first subtype of actions resulting from hand movements, general activities such as writing, typing, brushing teeth, clapping, folding clothes, playing with a tennis ball, etc. are dealt with; in the second subtype, the focus is, rather, on activities concerning eating such as drinking, eating sandwiches, pasta, chips, and so on. Data from smartwatches and smartphones are considered separately for each classifier and divided

into training, validation, and test sets according to a 60-20-20 scheme. The best results are achieved by CNN, which has good accuracy levels of 96.54%, the precision of 96.35%, and recall of 96.61% for data from a smartwatch. These results drop to about 90% for those obtained from smartphones. Changing classifiers, with the DeepConv LSTM, accuracies of between 87 and 88% are obtained for data derived from smartwatches, dropping to 75% for those obtained from smartphones.

Continuing along the lines of studies not purely inherent to the recognition of bullying or violence more generally, the study by Lee et al. [10] also proposes a distinction between three classes of activity with two classifiers, Random Forest and CNN from raw data and vector magnitude. This is referred to as a priori Feature Extraction in this case, which was probably carried out based on previous studies or knowledge. This study, however, prepares a simpler purpose than others by distinguishing between simple discernment activities such as walking, running, and standing still. The real purpose of the study is to show that, although the classification task is simple, Convolutional Neural Network manages to beat in performance (accuracy of 92.71%) classical machine learning algorithms such as Random Forest (accuracy of 89.10%).

Jordao et al. [11] propose the recognition of 12 everyday life actions through the sensors of a smartwatch placed at a sampling rate of 32 Hz. This low frequency led to low performance and an increase in the above could lead to a much better result than the 79.31% accuracy obtained with the CNN classifier at the end of the experiment. This study is interesting because it asserts that a low frequency of data could reasonably lead to an inaccurate representation of activity, which would not be the case with high frequencies.

Ismail et al. [12] propose a classification with the same number of classes as in this thesis, namely five, consisting of walking, sitting, standing up, climbing, and descending stairs. The data are hand-balanced and split into training and test sets according to an 80–20% instance-based proportion. Two optimizers are tested in the study, Adam, and RMSprop, and the latter performed best, coming in at 95.83% accuracy.

Many interesting studies have been analyzed regarding feature extraction, selection, and classification. It is useful, however, to dwell also on the type of pre-processing used to understand the State of the Art. In the related studies seen so far, all adopted the sliding window mechanism for data extraction, with the window size varying from one second to a maximum of nine seconds in the study by Cho et al. [3]. In these studies, the overlap, where implemented, ranged from 33% to 60%.

An interesting example from the perspective of sliding window overlap is that provided by the study of Dehkordi et al. [13], which uses a two-second sliding window with 75% overlap. The study is assumed to classify between two types of actions: static actions in which there is no real change in the global coordinates of the body and dynamic actions in which the whole body is in motion. Ten users were given a smartphone and instructed on what actions to perform with this smartphone held in their preferred hand. Data were collected at a frequency of 50 Hz from the triaxial accelerometer of the smartphones (a Samsung Galaxy or an LG Nexus). Many classifiers were implemented in the classification phase, including a Support Vector Machine, Decision Tree, and Naïve Bayes. The last two performed the best, reaching 98% average accuracy between static and dynamic task recognition.

*Studies Regarding Recognition of Bullying or Violence*

Studies of bullying-centered, sensor-based activity recognition are few, and most can be traced to a small circle of authors [14–17]. Given the small number of these studies, all the proposed approaches are reviewed below, and, in addition, those approaches involving fall recognition have also been considered, as this is one of the two actions that are recognized in this study as an action resulting from probable bullying activity. As previously stated, these articles belong to the same authors, therefore, we will examine how their approach has varied over time, starting, and continuing, in chronological order.

In 2014 Ye et al. [14] proposed a model that distinguishes everyday actions, such as walking, running, and bending-over, from bullying actions, such as pushing, hitting, and shaking, through a Fuzzy multiple-threshold classifier. The dataset is created with a roleplay phase with the help of some students who were made to wear inertial measurement sensors containing a triaxial accelerometer and triaxial gyroscope. Once the data is collected, the sliding window applied is eight seconds, with a 50% overlap, thus the sliding window is four seconds. Although the classification results are promising (accuracy of 92%), the discernment results of the crucial tasks, i.e., bullying tasks, are the worst with a classification error in the worst case, falling, of 25%.

Turning to 2015, Ye et al. [15] use the same set of features already defined in 2014 for activity recognition but add some activities more appropriate to the school case, such as playing or getting up from a chair and removing others that are not appropriate such as bending. Performance, however, drops to 80%, and cases of bullying actions misinterpreted go up. Activities such as shaking and colliding are classified as bullying actions with 58% and 69% accuracy, respectively, while falling from pushing remains stable with an error rate of 23%.

After three years, Ye et al. propose a new study [16] that, in addition to recognizing the movements of user subjects through sensor data, combines this decision with emotion recognition through analysis of an audio source. This study shows that the combination of these two methods works better than single methods; however, this study does not give much insight as far as our work is concerned. Ye et al.'s 2020 study [17] have the same structure; only the classifier, which has become a Radial Basis Function Neural Network, is changed. Also removed are two of the bullying tasks that gave the most problems, namely shaking and collision, replaced with pushing and shoving on the ground.

Another study dealing with Bullying Detection is that of Zihan et al. [18] which, like the previously cited paper by Ye et al. [15], combines Sensor-Based Activity Recognition and Emotion-Based Recognition of Tone of Voice. Sensor-Based recognition in this study was implemented with smartphone and smartwatch sensors. Some features were extracted a priori and then selected with the Principal Component Analysis algorithm. This is a dimensionality-reduction algorithm that takes feature spaces of size greater than or equal to four and recognizes which among them are the most discriminating for the classification task. For emotion recognition by voice, features were computed using the Mel Frequency Cepstral Coefficient (MFCC). This is a representation of the short-term power spectrum of a sound. The classifier used is K-Nearest Neighbors, and after cross-validation, the results were 77.8% accuracy for the Sensor-Based system and 81.4% accuracy for the Audio-Based system.

Finally, we consider studies using publicly available datasets found, both to take cues regarding the creation of our dataset and to see how they are used in the literature.

An example of the use of *UMAFall* comes from the study by Amara et al. [19]. In this paper, in addition to *UMAFall*, *SisFall* is also used, the latter being a dataset created with the participation of 38 volunteer subjects, divided into two groups based on age. The division of the data corresponds to 50% for training, 30% for validation, and 20% for testing. In pre-processing, however, a label is assigned to activities to identify whether they are Activities of Daily Living or falls. The classification is, therefore, binary, and the number of class instances is unbalanced. This class-unbalanced approach that results in binary classification is the only way *UMAFall* is used in studies purely related to Fall Detection. A confirmation of this statement can be found in the above-mentioned study by Cho et al. [2], who also discuss the work using *UniMiB SHAR*, saying that precisely because of the binary classification there is no real guideline for analyzing the results of their experiment. This problem comes up again, unfortunately, when we analyze the performance of our models.

Many studies have been conducted on falls, including that of Nguyen et al. [20]. In this paper, a method is presented that extracts 44 features on the domain: frequencies, time, and Hjorth parameters. The classifiers used were SVM, k-NN, ANN, J48, and RF. Following the F1-Score metric with the *MobileAct 2.0* dataset, the following values were found to be

95.23% (falls) and 99.11% (non-falls). In contrast, following the F1-Score metric with the *UP-Fall* dataset, 96.16% (falls) and 99.90% (non-falls) were found.

In this work, a comparison with the state of the art was made only for the Human Activity Recognition methodology, comparing values with known algorithms and a dataset like the one created by our team. Our results of the HAR macro-activities are in line with the current state of the art. The results inherent to cyberbullying/bullying are new, and we cannot currently compare them; therefore, we believe the current findings are a good starting point for other researchers to extend the work.

### 3. Dataset

The datasets considered belong to two categories. The first category considers the two HAR datasets "*DatasetUniba*" and "*UniMiB SHAR*", which have the classical feature columns that identify a HAR Dataset with an accelerometer; thus, the raw x, y, z, and target features of the activity. While the second category "*Questionnaire experiment dataset*" identifies the dataset extracted from the user test with the questionnaire. This dataset has three accelerometer features (x, y, z). In addition, ideally, one more feature was considered for the final analysis, namely, the users' personality index (*Bully*, *Cyberbully*, *Bully Victim*, *Cyberbully Victim*) provided as output from the questionnaire.

*DatasetUniba:*

The creation of this dataset took place in a controlled environment. Nineteen participants, thirteen males, and six females took part in the construction.

Each was made to perform eight actions divided into the two categories already encountered:

- *ADLs* (walking, running, jumping, sitting);
- *Falling* (forward, backward, right, and left).

A smartphone was placed in the right pocket with the screen toward the participants' bodies, and accelerometer values were collected at a sampling rate of 200 Hz via the smartphone application. The data were sent to a server in .txt format and were then reprocessed to arrive at the final format. Each task was performed between two and three times. Each trial was 15 s long. The falls were performed on a mattress placed on the floor. Within the .csv file, the first column contains the user id, the second the activity performed, the third contains the timestamp in milliseconds from the start of the action, and next are the three triaxial accelerometer values x, y, and z.

*UniMiB SHAR:*

A Dataset consisting only of values obtained from the accelerometer. Thirty participants. Recognized activities are again divided into two categories:

- *Falling* (Falling forward, backward, right, left, hitting an obstacle, with protective strategies, backward without protective strategies, Syncope)
- *ADL* (Walking, running, climbing stairs, descending stairs, jumping, lying down from standing, sitting)

For each activity, there are two to six trials for each user. For the actions with two trials, the smartphone in the right pocket is used in the first and the smartphone in the left pocket in the second. For actions with six trials, the first three have the smartphone in the right pocket and the other three in the left pocket. Data are provided in windows of 51 or 151 samples around an original signal peak higher than 1.5 g, with g being the acceleration of gravity [5].

*Questionnaire experiment dataset:*

The *Questionnaire experiment test* consists of an Android application that incorporates a questionnaire designed by psychologists. The graphical layout of the existing app is simple and essential. Every choice has been made to make the graphical interface simple and appealing to the end-user and, to eliminate, or at least drastically reduce, the possibility that the user during the activities may make mistakes may have doubts about the actions to be performed, or other problems in general. The application contemplates five basic steps:

1. Access to the app's functionality (3 screens);
2. Access to the four Cyberbullying videos (4 screens) for each video an emotional question (4 screens);
3. Access to the Questionnaire (30+ screens) that is quick to fill out;
4. Access to the Telegram group (1 screen);
5. End of Test (1 screen).

This specific test returns the personality index: *Bully*, *Cyberbully*, *Bully Victim*, *Cyberbully Victim* for each user [21].

The four Cyberbullying videos specifically are [21]:

1. *Video1Activity* ("*VIRTUAL ACTIONS, REAL CONSEQUENCES*", https://www.youtube.com/watch?v=x2AxcllGLJg&t=4s&ab_channel=TabbyEUproject, accessed on 3 January 2023. [21])
2. *Video2Activity* ("*ANYONE CAN BE ANYONE*", https://www.youtube.com/watch?v=z3N24DpD64c, accessed on 3 January 2023. [21])
3. *Video3Activity* ("*INTERNET = EVERYONE, FOREVER*", https://www.youtube.com/watch?v=K31Kuc5pTXM&t=42s&ab_channel=TabbyEUproject, accessed on 3 January 2023. [21])
4. *Video4Activity* ("*VIRTUAL VENDETTA (joke or* crime?)", https://www.youtube.com/watch?v=FpBVBwv6UQ4&ab_channel=TabbyEUproject, accessed on 3 January 2023. [21])

The dataset used in this study is the result of a *Questionnaire experiment test* using an Android application created and used to acquire data. The tests were conducted with 99 students aged 18–24, (average age 20) all enrolled in their first year of college. There were two test sessions carried out on real users (referred to in the paper as Test 1 and Test 2). The users in the two tests were different. Test 1 and Test 2 were set up by allowing participants to sit at their desks. Allowing them any freedom to move around. However, the results predicted sedentary activity from the accelerometer readings taken during the questionnaire. Sitting is defined as the baseline activity in the paper. Sitting was then considered non-restless behavior, and, therefore, quiet.

## 4. Experimental Setup

The Experimental Setup was structured into several parts:

- *Phase One*: Extraction of sensor coordinates;
- *Phase Two*: Model training;
- *Third Phase*: Recognition of the activity performed by the model above.

Phase One is devoted to the extraction of the *Raw Accelerometer Coordinates (X, Y, Z)* from the experiment performed in real-world contexts with the questionnaire application. The accelerometer was recorded for each screen of the application. Each screen included a single question from the questionnaire application. The resulting Dataset from the extraction, for each user of the 90 tested, is used in the final stage as a test with the HAR model. In Phase Two, the HAR model is trained with two HAR datasets, one state-of-the-art "*UniMiB SHAR* [5]" and the second created by our research group "*DatasetUniba*". In the Third Phase, through the HAR model, for each user, the activities are predicted using the experiment of the first phase.

### 4.1. Data Preprocessing

At this phase, text files containing data obtained from device sensors from the questionnaire app were retrieved, but only the accelerometer data was retained, and those from the magnetometer and gyroscope were discarded. Once the researcher selected the row of interest (*with X, Y, Z coordinates*), they would insert a row for each triad of coordinates (*X, Y, Z*) within a data structure, which also specified the recorded questionnaire screen during that movement. CSV files were created for each user, which was helpful in later stages for predictions. The same procedure was followed for both the users participating in

Test 1 (52) and Test 2 (48), of which, however, not all users were considered because some did not complete the entire questionnaire.

*4.2. Models*

State-of-the-art models were considered for this test:

- *CNN*, a convolutional, feed-forward neural network, consisting in this case of 3 ReLU layers, each alternating with a pooling layer for simplifying the output obtained from the previous layer, whose practical goal is to reduce the number of parameters the network must learn. After these, a Flatten layer is for linearizing the output and a *softmax* layer is for actual classification;
- *Bi-LSTM*, this model is a special type of LSTM network that is practically trained to make predictions based not only on knowledge of the past but also the future and then go backward with the predictions. This is unlike the LSTM network, which can learn only unidirectionally.

Table 1 shows the parameters used by the networks for this work.

**Table 1.** Parameters used for networks.

| CNN | Bi-LSTM |
| :---: | :---: |
| Model Sequential | Model Sequential |
| Conv2D: filters 64, kernel_size 2, activation relu | Layers LSTM: units 64 |
| Dropout 0.5 | Dropout 0.2 |
| Conv2D: filters 32, kernel_size 2, activation relu | Layers LSTM: units 64 |
| Dropout 0.5 | Dropout 0.2 |
| Conv2D: filters 16, kernel_size 2, activation relu | - |
| Flatten layer | - |
| Dense layer 256, activation relu | Dense layer 64, activation relu |
| Dropout 0.5 | Dropout 0.5 |
| Optimizer adam, loss categorical_crossentropy | Optimizer adam, loss categorical_crossentropy |

## 5. Results

We employed several Deep Learning models with different combinations for the HAR experimentation to find the model that offered the best performance. Table 2 shows the averages of accuracy and f1 score overall users. Specifically, they constitute the averages of the averages concerning each user in the Dataset trained on using the *Leave One Out technique*.

**Table 2.** Best accuracy results and average F1 scores.

| Model | Dataset | Average Accuracy | Average F1_Score |
| :---: | :---: | :---: | :---: |
| CNN | *DatasetUniba* | *0.9199* | *0.9084* |
| CNN | *UnimibShar* | *0.9768* | *0.9473* |
| Bi-LSTM | DatasetUniba | 0.8955 | 0.8581 |
| Bi-LSTM | UnimibShar | 0.7617 | 0.7885 |

In Table 2, it can be observed that the CNN model outperforms the Bi-LSTM models. This means that the model could better discriminate all activities from the *DatasetUniba* and *UnimibShar datasets*, even similar activities such as falling and sitting. Thus, CNN, as the best-performing model, was used to perform activity prediction with the dataset "*Dataset questionnaire experiment*" to analyze activity predictions by class. The results by classes of interest are illustrated in Figures 1–4. In Figures 1–4, the color legend in the graphs is as follows: Bullying (Gray), Victims of bullying (Blue), cyberbullying (Orange), and Victims of Cyberbullying (water green).

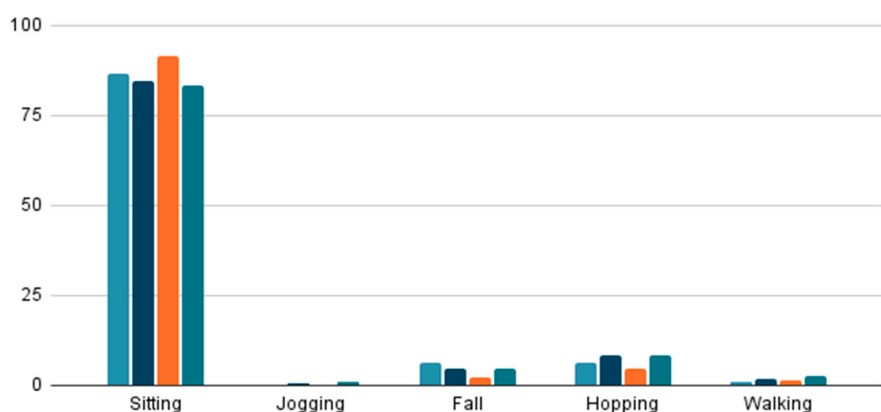

**Figure 1.** Averages Results in Users Test 1. Bullying (Gray), Victims of bullying (Blue), cyberbullying (Orange), Victims of Cyberbullying (water green). The x-axis identifies activities, while the y-axis identifies the average percentage of activities, of each test participant, divided by personality index (questionnaire class).

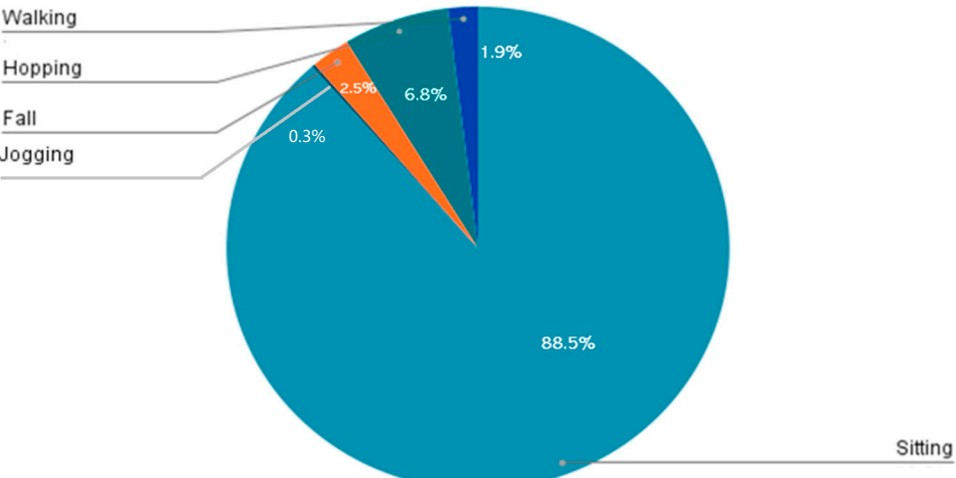

**Figure 2.** Total-User Averages Test 1: Bullying (Gray), Victims of bullying (Blue), Cyberbully (Orange), Victims of Cyberbullying (water green).

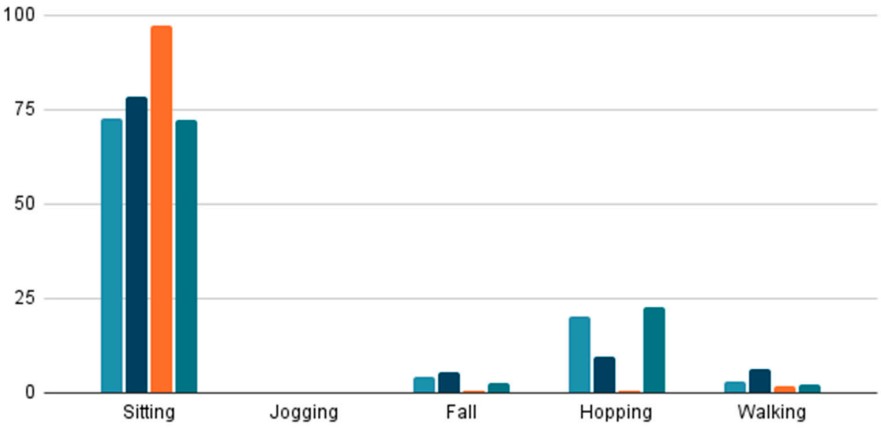

**Figure 3.** Averages Results in Users Test 2. Bullying (Gray), Victims of bullying (Blue), cyberbullying (Orange), Victims of Cyberbullying (water green). The x-axis identifies activities, while the y-axis identifies the average percentage of activities, of each test participant, divided by personality index (questionnaire class).

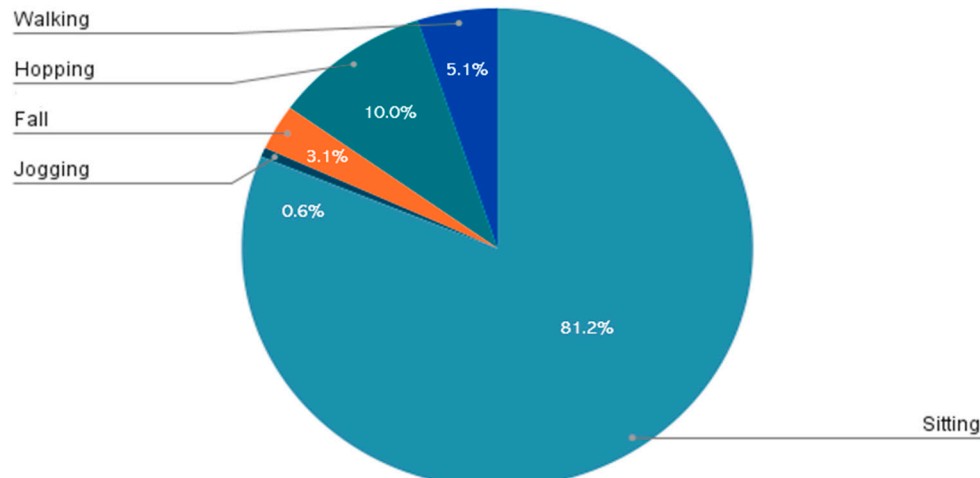

**Figure 4.** Total-User Averages Test 2. Bullying (Gray), Victims of bullying (Blue), cyberbullying (Orange), Victims of Cyberbullying (water green).

Regarding Figures 1 and 2, the y-axis identifies the average percentage of activities, of each individual participating in the individual test, divided by personality index (questionnaire class). The individual has his or her percentage of activity about filling out the questionnaire and, in turn, possesses a personality index. It should be noted that these percentages derived from each individual are averaged.

Looking at the results obtained from the experiment, it can be seen that (Figures 1–4), considering Test 1 and Test 2:

- *Cyberbullying* was found to be "quieter" because users were recognized as sitting most of the time and in higher percentages than the other categories of users;
- Bullying has jumped as their predominant abnormal activity, particularly during the questionnaire phases of *QuizActivityButtons*. The *QuizActivityButton* stage contemplates questions that ask about activities performed or experienced in Bullying and Cyberbullying;
- Bullying victims, as well as Bullying, showed more abnormal phases during the *QuizActivityButton* but with prevalent activities of jumping, walking, and falling. Falling is also defined as dropping the cell phone;
- The victims of *Cyberbullying* were the only users participating in the questionnaire who showed several abnormalities in the initial part, in which they were subjected to some videos to watch with questions related to the feelings experienced by watching them.

It is believed that the questionnaire could be reduced to only those questions about the category it is intended to identify:

- In the case of Bullying and Victims of Bullying, *QuizActivityButtons* would suffice, thus removing the initial part of the video submission and related questions;
- The latter, however, was useful for the category of cyberbullying victims with additional questions reported below:
  - *Video1Activity* ("*VIRTUAL ACTIONS, REAL CONSEQUENCES*", https://www.youtube.com/watch?v=x2AxcllGLJg&t=4s&ab_channel=TabbyEUproject, e.g., accessed on 3 January 2023. [21])
  - *Question_Video1Activity (5-point Likert scale emotional question);*
  - *Video2Activity* ("*ANYONE CAN BE ANYONE*", https://www.youtube.com/watch?v=z3N24DpD64c, accessed on 3 January 2023. [21])
  - *Question_Video2Activity (5-point Likert scale emotional question);*
  - *Video3Activity* ("*INTERNET = ALL, FOREVER*", https://www.youtube.com/watch?v=K31Kuc5pTXM&t=42s&ab_channel=TabbyEUproject, accessed on 3 January 2023. [21])

- *Question_Video3Activity (5-point Likert scale emotional question);*
- *Video4Activity ("VIRTUAL SALVAGE (joke or crime?)", https://www.youtube.com/watch?v=FpBVBwv6UQ4&ab_channel=TabbyEUproject, accessed on 3 January 2023. [21])*
- *Question_Video4Activity (5-point Likert scale emotional question);*
- *Activity_Curtain: Select from the drop-down menu the plexus to which you belong.*
- *QuizActivityText 0: Enter the modified Telegram Nickname at the beginning of the app.*
- *QuizActivityText 1: How old are you?*
- *QuizActivityText 2: Are you a boy or a girl?*
- *QuizActivityText 3: What school do you attend?*
- *QuizActivityText 4: What grade do you attend?*
- *QuizActivityText 5: What country were you born in?*
- *QuizActivityText 6: Specify what part of Italy you live in.*
- *QuizActivityText 7: Write in the box below, What social networks do you use?*
- *QuizActivityText 8: If you have at least one profile, how many friends do you have on social networks?*
- *QuizActivityText 9: Is one of your parents or another adult you trust among your friendship contacts? (YES/NO)*
- *QuizActivityText 10: Do you have at least one profile on social networking sites (Example: Facebook, WhatsApp, Instagram, Ask.com, etc.)?*

Looking at the results and the questions, it is possible that the victims of Cyberbullying were strongly influenced and disturbed by the videos containing images of Bullying but especially Cyberbullying. It could be that some Cyberbullying victims may have been disturbed by remembering some events that happened in the past. In addition, abnormal activities were also recorded in the generic initial questions, this is because victims always tend to hide and not be found out.

- Regarding Cyberbullying, *QuizActivityButtons* 55 to 59 were found to be more discriminating, particularly with Falling activities:
  - Receiving threats and insults on the Internet (websites, chat rooms, blogs, SMS, Facebook, Twitter . . . ),
  - Receiving silent phone calls
  - Receiving e-mails with threats and insults
  - Receiving videos/photos/pictures of embarrassing or intimate situations via cell phone
  - Receiving phone calls with threats and insults

Looking at the results and the questions, it is possible that Cyberbullying is highly conditioned by their actions; therefore, contentment about the victim's receipt of content could be weighed positively and, thus, appear to be discriminated against.

## 6. Conclusions

In this study, through *Human Activity Recognition (HAR)* models, behavioral analysis was performed by analyzing users' behavior while filling out a questionnaire useful for classifying users as *Bullying*, *Cyberbullying*, *victims of bullying*, and *victims of Cyberbullying* via a smartphone. The smartphone questionnaire incorporated accelerometer data useful for recognizing activities and behaviors different from just sitting. Any activity other than sitting (considered non-abnormal) is classified as abnormal or representative. Questions that coincided with activities classified as "Abnormal" were considered for analysis among the Target classes.

Observation and conclusions:

- Appropriate DL models could be used to perform HAR based on data obtained from sensors on a smartphone, and the best-performing Deep Learning model was the convolutional neural network;

- Abnormal aspects were observed and highlighted during the prediction phase itself, in the form of activity different from what is expected (sitting). Cyberbullying was found to be "quieter" because users were recognized as sitting most of the time. Bullying is noted as an abnormal activity in the form of jumping during the steps of the *QuizActivityButton* questionnaire, which contemplates questions asking about activities performed or experienced in Bullying and Cyberbullying. Bullying victims and Bullying showed more abnormal activity during the *QuizActivityButtons* with prevalent activities of jumping, walking, and falling. The victims of Cyberbullying were the only users participating in the questionnaire who demonstrated several anomalies concerning all activities in the HAR datasets.
- For each category, "*anomalous*" questions were selected and reported in the paper justified by appropriate theories inherent in the various classifications.

There are many limitations to the current study: one problem may be the small amount of data. Both the HAR dataset and the dataset resulting from the questionnaire test contain small numbers of users. In addition, a major limitation is that we are proposing a methodology that brings together HAR and cyberbullying/bullying via smartphone sensors, which has not yet been widely addressed in the state of the art. Plus, it might be interesting in the future to try multiple other algorithms that can see the micro-behaviors identified by the personality index.

The results inherent to HAR are in line with the state of the art even considering a second dataset namely the "*DatasetUniba*". While the results derived from the questionnaire study and the marriage with the HAR approach are embryonic and can be considered as a pathfinder for future work. However, while embryonic, the conclusions made are still interesting because correlations were found between the psychology and behavioral attitude of the individual.

**7. Patents**

This section is not mandatory but may be added if there are patents resulting from the work reported in this manuscript.

**Author Contributions:** Conceptualization, V.G., D.I., G.P. and L.S.; methodology V.G., D.I., G.P. and L.S.; software, V.G.; validation, V.G.; investigation, V.G., D.I., G.P. and L.S.; data curation, V.G.; writing—original draft preparation, V.G.; writing—review and editing, V.G., D.I., G.P. and L.S.; visualization, V.G., D.I., G.P. and L.S.; supervision, D.I. and G.P.; project administration, D.I.; All authors have read and agreed to the published version of the manuscript.

**Funding:** This work is funded by the Italian Ministry of Education, University, and Research within the PRIN2017—BullyBuster project—A framework for bullying and cyberbullying action detection by computer vision and artificial intelligence methods and algorithms. CUP: H94I19000230006.

**Institutional Review Board Statement:** Not applicable.

**Informed Consent Statement:** Not applicable.

**Data Availability Statement:** The study did not report any data.

**Acknowledgments:** This work is supported by the Italian Ministry of Education, University, and Research within the PRIN2017—BullyBuster project—A framework for bullying and cyberbullying action detection by computer vision and artificial intelligence methods and algorithms. All individuals included in this section have consented to the acknowledgment.

**Conflicts of Interest:** The authors declare no conflict of interest.

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
