# Peer review of "Human Activity Recognition for the Identification of Bullying and Cyberbullying Using Smartphone Sensors"

_electronics, doi:10.3390/electronics12020261_

Round 1

Reviewer 1 Report

1) The problem statement/need for the proposed methodology is not mentioned.

2) The drawbacks of the earlier works are not clearly mentioned in the survey section

3) Methodology part is very weak. No scientific methods are used. Is it only a study experiment? If yes, other journals may be more suitable.

4) Are the sensors used in this work developed by the authors? If yes, more information must be given on it.

5) Cyber bullying deals predominantly with computers/informatics. No such information is given.

6) The results do not match with the objective of the work.

7) There is no concrete validation on the results

8) It is more suited for a psychology/education related journal

Author Response

Reviewer 1, Please see the attachment

Reviewer 2 Report

The authors have developed a model that can distinguish between behaviors taken in daily life and those brought on by suspected bullying activities using the most recent analysis of HAR. Walking, jumping, sitting, running, and falling are the five acts they have taken into consideration for recognition. In order to conduct the analysis, the dataset from the smartphone questionnaire application experiment was subjected to the best HAR activity identification model. Their approach looks fresh and has been executed flawlessly technically, making it worthwhile to publish. However, the paper has to be proofread because it has a number of grammatical errors. The abstract is well written. The themes are meaningful, insightful, topical, and helpful for future research.

Author Response

For Reviewer 2, Please see the attachment

Reviewer 3 Report

This paper presents a recognition method to identify the human activities of bullying and cyberbullying using smartphone sensors. The topic is very important for the next generation since social media has already become the main communication tool among teenagers. However, this paper needs to be improved according to the following comments.

1.     No figure or table to demonstrate sample data from the data set used in this research. Please provide at least one example of bullying and cyberbullying activities. 

2.     No figure or table shows the network structure used in this research. Please provide more details about the network structure, such as the layer numbers or the hyperparameters that the authors decided.

3.     What is the y-axis indicating in Figure 1? Please add the label of the y-axis. The reviewers could not understand why the percentage of sitting is so high than others, in Figure 1 and Figure 2. Please present more discussions on this issue.

4.     The originality of this research should be more stressed by comparing it with previous research.

5.     The limitation of this research needs to be discussed.

6.     Please discuss how the result obtained from this research can be used in future work.

Author Response

For Reviewer 3, Please see the attachment.

Round 2

Reviewer 1 Report

It can be accepted now 

Reviewer 3 Report

The authors have responded to the reviews and improved the article.